# Exploring the Functional Properties of Propolis, Geopropolis, and Cerumen, with a Special Emphasis on Their Antimicrobial Effects

**DOI:** 10.3390/foods12213909

**Published:** 2023-10-25

**Authors:** Bajaree Chuttong, Kaiyang Lim, Pichet Praphawilai, Khanchai Danmek, Jakkrawut Maitip, Patricia Vit, Ming-Cheng Wu, Sampat Ghosh, Chuleui Jung, Michael Burgett, Surat Hongsibsong

**Affiliations:** 1Meliponini and Apini Research Laboratory, Department of Entomology and Plant Pathology, Faculty of Agriculture, Chiang Mai University, Chiang Mai 50200, Thailand; pichet.p@cmu.ac.th (P.P.); michael.burgett@oregonstate.edu (M.B.); 2ES-TA Technology Pte Ltd., Singapore 368819, Singapore; kaiyang.lim.john@gmail.com; 3Office of Research Administration, Chiang Mai University, Chiang Mai 50200, Thailand; 4School of Agriculture and Natural Resources, University of Phayao, Phayao 56000, Thailand; khanchai.da@up.ac.th; 5Faculty of Science, Energy and Environment, King Mongkut’s University of Technology North Bangkok, Rayong Campus, Bankhai, Rayong 21120, Thailand; jakkrawut.m@sciee.kmutnb.ac.th; 6Apitherapy and Bioactivity, Food Science Department, Faculty of Pharmacy and Bioanalysis, Universidad de Los Andes, Merida 5001, Venezuela; vitolivier@gmail.com; 7Department of Entomology, College of Agriculture and Natural Resources, National Chung Hsing University, Taichung 40227, Taiwan; mcwu@nchu.edu.tw; 8Agriculture Science and Technology Research Institute, Andong National University, Andong 36729, Republic of Korea; sampatghosh.bee@gmail.com; 9Department of Plant Medical, Andong National University, Andong 36729, Republic of Korea; cjung@andong.ac.kr; 10Department of Horticulture, Oregon State University, Corvallis, OR 97331, USA; 11School of Health Sciences Research, Research Institute for Health Sciences, Chiang Mai University, Chiang Mai 50200, Thailand

**Keywords:** antimicrobial potency, antibiotics replacement, cerumen, geopropolis, honey bee, propolis, quality factors, stingless bee

## Abstract

Bee propolis has been touted as a natural antimicrobial agent with the potential to replace antibiotics. Numerous reports and reviews have highlighted the functionalities and applications of the natural compound. Despite much clamor for the downstream application of propolis, there remain many grounds to cover, especially in the upstream production, and factors affecting the quality of the propolis. Moreover, geopropolis and cerumen, akin to propolis, hold promise for diverse human applications, yet their benefits and intricate manufacturing processes remain subjects of intensive research. Specialized cement bees are pivotal in gathering and transporting plant resins from suitable sources to their nests. Contrary to common belief, these resins are directly applied within the hive, smoothed out by cement bees, and blended with beeswax and trace components to create raw propolis. Beekeepers subsequently harvest and perform the extraction of the raw propolis to form the final propolis extract that is sold on the market. As a result of the production process, intrinsic and extrinsic factors, such as botanical origins, bee species, and the extraction process, have a direct impact on the quality of the final propolis extract. Towards the end of this paper, a section is dedicated to highlighting the antimicrobial potency of propolis extract.

## 1. Introduction

### 1.1. Propolis and Geopropolis

Propolis, often referred to as “bee glue”, is a sticky substance that bees gather from plant exudates and buds. They blend this material with their own mandibular secretions, enzymes, wax, and plant resin [1,2]. Honey bees (*Apis mellifera*) create propolis, and its composition typically includes around 50% plant resins, 30% waxes, 10% essential and aromatic oils, 5% pollens, and 5% other organic compounds [3,4]. The effectiveness of different types of propolis is closely tied to their complex chemical makeup, which can differ based on factors, like the season, the location from where plant resins are collected, and the species of bee [5]. The term “propolis” originated from two ancient Greek words: “pro”, which signifies before or protection, and “polis”, which means city [6]. Propolis serves as a significant defensive material for bee nests due to its adhesive and resinous nature [7]. As a result, bees utilize it to secure and reinforce their hives [8,9]. Beyond functioning as a construction glue, propolis also has a vital role in maintaining colony health due to its antimicrobial and antifungal properties [10,11].

The abundant therapeutic potential of propolis has led to its incorporation as a bioactive element in traditional remedies [12]. The earliest recorded use of propolis in traditional folk medicine dates to ancient Egypt in 3100 BC, where it was considered an elixir presenting eternal health and life. Even in contemporary times, bee propolis is marketed and consumed as a highly valuable nutraceutical product, offering diverse health advantages. Propolis boasts an array of biological and pharmacological properties, encompassing antimicrobial, antifungal, anti-ulcer, anticancer, anti-parasitic, antioxidant, antiviral, anti-inflammatory, and wound-healing traits [13,14,15,16,17,18,19,20,21,22]. Most of the research documented in the international literature revolves around propolis collected by *A. mellifera*, while propolis sourced from other bee species, including Meliponini, has garnered less attention.

Geopropolis is a distinctive kind of propolis created by stingless bees (Meliponini). It is formulated by blending plant resins, waxes, and earth debris. Unlike the propolis generated by honey bees, geopropolis incorporates wax and soil into its composition, resulting in unique attributes. Despite its importance, the geopropolis collected by stingless bees has received limited attention and is not extensively detailed in the available literature [5,23,24,25]. Geopropolis has less resinous material and plant trichomes than *A. mellifera* propolis, but it includes more minerals and soil. Geopropolis performs the same function as propolis in the nest by combining plant resins with waxes and earth [24,25]. Stingless bees exclusively inhabit tropical and subtropical regions, comprising a diverse group of over 605 species [26].

### 1.2. Cerumen

The cerumen of stingless bees refers to the wax-like material produced by the worker bees of the stingless bee species belonging to the tribe Meliponini. Cerumen is a mixture similar to propolis, and stingless bees produce it by combining plant resins with waxes. There is suggestive evidence indicating that during cerumen production, stingless bees introduce secretions from their head glands [27]. Nevertheless, as outlined by Spivak [8], the combination of plant resins with soil or clay substances gives rise to a mixture termed geopropolis or batumen. Conversely, when resins are exclusively blended with wax, it is referred to as cerumen, specifically in the context of non-honey bee species [28]. Stingless bees produce two distinct substances used for nest construction and nest maintenance called cerumen and propolis. While the terms “cerumen” and “propolis” are sometimes used interchangeably in the literature concerning stingless bees, they refer to separate materials with unique functions [29]. Unlike honey bees, which employ propolis for the interior lining around nest combs, stingless bees primarily use cerumen as a building material for various nest elements, particularly for building brood cells, involucrum, pillars, nest entrances, sealing storage pots (honey pots and pollen pots), and other structures within the nests. It has sometimes been called batumen [30,31,32]. In the nest, cerumen serves various purposes, such as mummifying intruders and maintaining a hygienic environment within the hive [29,33]. It plays a vital role in the protection and maintenance of their colonies [34].

In Thailand, around 40 species of stingless bees have been identified to date. Among these, *Tetragonula laeviceps* stands out as one of the most prevalent stingless bee species in Thailand. This bee species plays a significant role in producing substantial quantities of honey and cerumen. Moreover, they are extensively employed in stingless bee keeping (meliponiculture) due to their adaptable nature for nesting in hollow or cavity structures [35]. Additionally, various other species from the *Tetragonula* genus, such as *T. fuscobalteata*, *T. pagdeni*, and *T. testaceitarsis*, are commonly utilized for meliponiculture throughout the country. Thailand also manages other bee species, like *Lepidotrigona flavibasis*, *L. doipaensis*, and *L. terminata* [36]. In the southern regions of Thailand, *Heterotrigona itama* and *Geniotrigona thoracica* have gained popularity as prominent species in the stingless bee industry.

### 1.3. Properties of Propolis, Geopropolis, and Cerumen

The extensive molecular composition of propolis, comprising as many as 300 constituents, contributes to its biological attributes [37,38]. Studies have indicated that the predominant factors behind the biological properties of propolis are mainly flavonoids, phenolic acids, terpenes, and sugars. These components are responsible for various propolis-associated qualities, such as antibacterial effects [39], anti-fungal actions, anticancer potential [40], anti-tumoral properties [41], anti-protozoal capabilities, anti-inflammatory responses, hepatoprotective benefits, antioxidative effects, as well as antiviral and antimicrobial properties [42]. Propolis exhibits various therapeutic attributes encompassing the management of conditions, like cancer, oral ailments, cardiovascular disorders, and wound healing. Its extensive utilization spans the realms of the food, veterinary, pharmaceutical, and cosmetic industries. In the domain of nutrition, propolis functions as a functional component, offering relief for throat discomfort, regulation of sugar intake, enhancement of the immune system, and augmentation of energy levels. However, propolis finds its way into food products mainly within the categories of confectionery, spreads, and pet foods. An inherent quality of propolis lies in its effective antimicrobial prowess, positioning it as a prospective alternative to prevailing gold standard antibiotics for addressing clinical microbial infections [12,43,44,45].

It is worth mentioning that the biological activities of geopropolis produced by stingless bees have been attributed to their phytochemical composition. The influence of the inorganic content (minerals, soil, or clay particles) or even organic material associated with geopropolis, such as native microbiota or decomposing organisms, has not been addressed in the literature. This review focuses on the chemical profile and biological effects (antioxidant capacity, antimicrobial, and toxic potentials) of geopropolis produced by stingless bees native to Brazil. Moreover, the major geopropolis components pointed out here are subjected to a toxicological analysis in order to provide additional evidence of their safe use. Different biological activities of geopropolis have been investigated worldwide, including antioxidant [25,46,47,48], anti-inflammatory [48,49,50], anti-biofilm [49], anticancer [51,52], antimicrobial activities [25], immunomodulatory effects, and toxicity [1,50,53]. Alves de Souza et al. [54] documented the composition and antioxidative potency of geopropolis sourced from *Melipona subnitida* (Jandaíra) bees. In certain nations, geopropolis has been traditionally employed by communities for wound healing, managing gastritis, and functioning as an antibacterial agent, as reported by Sawaya, Barbosa da Silva Cunha and Marcucci [55]. The investigation into the properties of propolis from stingless bees in Thailand is limited. Sanpa et al. [39] outlined the chemical constitution and antimicrobial effects of propolis obtained from two stingless bee species: *T. laeviceps* and *Tetrigona melanoleuca*. Umthong, Puthong and Chanchao [56] observed the antimicrobial, antiproliferative, and cytotoxic capacities of *T. laeviceps* propolis, along with the in vitro antiproliferative potential of partially purified *T. laeviceps* propolis from Thailand on human cancer cell lines. The fungicidal properties of geopropolis have also been explored.

The misuse of antibiotics and the dearth of novel antibiotic developments have accelerated the emergence of antibiotic resistance among virulent microorganisms. At present, approximately 50,000 deaths are attributed to infections caused by antimicrobial-resistant strains in Europe and the United States [57]. This figure is projected to surge by twenty-fold within the next three decades [58,59]. Within the medical community, there is an urgent request for an alternative antimicrobial agent that can effectively take the place of the diminishing effectiveness of antibiotics. Bee propolis emerges as a prospective challenger due to its potent and broad-spectrum antimicrobial capacity [60,61,62]. The fact that the compound is derived naturally enhances the public’s receptivity and acceptance of its clinical application. This paper provides a comprehensive review of the general physicochemical characteristics of propolis and the natural synthesis of propolis by honey bees, as well as evaluates the effect of botanical origin, bee species, and geographical location on the physical and chemical characteristics and biofunctional properties of propolis. A special section, towards the end of the review, is dedicated to critically evaluating the biotherapeutic potential of bee propolis. This provides an insight into the possibility of using propolis as a next-generation bioactive compound for biomedical applications.

The investigation into cerumen properties remains quite limited in the existing literature. Pérez-Pérez et al. [63] compared phytochemicals and antioxidant activity in cerumen honey pots, involucrum of the brood, entrance tube, and propolis of a *Tetragonisca angustula* (Latreille, 1811) nest in Merida, Venezuela. The major flavonoid and protein contents were found in propolis, whereas the honey pots had a major polyphenol content and antioxidant activity in these nest materials. According to Massaro et al. [34], extracts derived from the cerumen of stingless bees exhibit an anti-inflammatory potential by impeding enzymes that catalyze the activity of inflammatory agents. These cerumen extracts demonstrate comparable effects to the positive control, trolox (a vitamin E-like antioxidant), albeit with lesser inhibitory potency than propolis from honey bees. Additionally, the extracts from stingless bee cerumen were explored for their potential as an anticancer agent against various human cancer cell lines, such as breast, lung, liver, stomach, and colon. These extracts induced considerable cytotoxicity and cell morphology reminiscent of apoptosis. Notably, this research highlighted that α-mangostin, extracted from cerumen, demonstrated in vitro cytotoxicity against the aforementioned cell lines and in vivo cytotoxicity against zebrafish embryos [64].

Propolis obtained from other bee species, such as Meliponini, has received less attention than propolis obtained from *A. mellifera* in the international literature. Therefore, we review the propolis produced by *A. mellifera* and Meliponini and the cerumen and geopropolis that are exclusive Meliponini nest products. Propolis is not a novel antibacterial agent, as informed by Almuhayawi [65], because it has been considered a powerful antibacterial agent for bees [8] and human health [7,66] since the initial studies. However, propolis applications are developing as novel functional foods and nutraceutical ingredients [67]. To achieve a comprehensive understanding of the strong antimicrobial attributes exhibited by propolis, geopropolis, and cerumen, an in-depth knowledge of these substances is imperative. This encompasses a profound understanding of their physical characteristics and chemical constituents. Furthermore, a thorough exploration of the intricate process of propolis synthesis by bees, the impact of botanical origins, bee species variations, and extraction methodologies on the efficacy of propolis as an antimicrobial agent, is essential. Moreover, it is critical to acknowledge the necessity for regulatory guidelines and standards aimed at protecting both consumers and producers against potential adulterations in the utilization of these natural materials for antimicrobial purposes.

Bee products have gained significant attention due to their multifaceted properties, encompassing not only nutritional value, but also profound physiological and biological effects. In this context, propolis, geopropolis, and cerumen, a resinous substance produced by bee species, stand out as remarkable natural materials with a wealth of physicochemical, biological, and nutritive properties.

## 2. Characteristics of Propolis

### 2.1. Physical Properties

Propolis emerges as a sticky, resin-like substance exhibiting a darkened shade. While the predominant form of documented propolis displays a deep-brown color, it also exhibits variations in shades, like green, red, or yellow. Cuesta-Rubio et al. [68] detailed the identification and characterization of propolis with a yellow hue, discovered in Cuba. In a separate account, the presence of red propolis in beehives situated along the sea and riverbanks of northeastern Brazil was emphasized [69]. The physical attributes of propolis are subject to fluctuations influenced by factors both internal and external, encompassing aspects, like age, bee species, plant resin origins, and geographical location [70]. Propolis exists as a waxy, pliable, and adhesive material under normal atmospheric conditions. To the best of our knowledge, no direct textural or rheological studies have been conducted on raw propolis due to the technical difficulties in sampling handling and preparation. A simple characterization study on the rheological properties of honey and propolis illustrates that pure propolis extract, dispersed in water or alcohol, is a Newtonian fluid, maintaining its viscoelastic properties with an increasing shear rate [71]. In fact, the consistency of propolis is affected by both intrinsic and extrinsic factors. An important intrinsic factor affecting the consistency of propolis is the content of beeswax incorporated within the material. A higher beeswax content often gives rise to a waxier viscoelastic consistency [72]. Temperature represents a crucial external factor that can directly affect the physical properties of propolis. When cooled below 15 °C, propolis freezes to a rigid and brittle crystalline solid. At the other extreme, propolis has a known melting point of approximately 65 °C; however, in some samples, it might go as high as 100 °C [4,10,73]. Above the melting point, propolis melts to form a viscous fluid. 

Propolis, generally, has a characteristically strong odor and aroma, similar to that of an aromatic gum resin [74]. In fact, propolis is widely used as a substitute for galbanum in consumer care products, such as perfumes. The odor of propolis is attributed to the presence of volatile compounds entrapped within the viscoelastic matrix. Such propolis volatiles are usually present in low concentrations, ranging from 0.02% to 3.00%. Bankova, Popova and Trusheva [75] conducted a comprehensive review of the volatile constituents found in propolis from various geographical origins. Sesquiterpenes, hemiterpene alcohols, oxygenated monoterpenes, and oxygenated aliphatic hydrocarbons represent the predominant volatile organic compounds (VOCs) found in propolis. These volatiles are responsible for the characteristic woody odor that most propolis possess.

### 2.2. Chemical Compositions

The complex and varied nature of propolis results in its complex chemical structure, with over 300 elements identified to date [76,77]. An extensive examination of numerous literature sources reveals that propolis is primarily composed of key constituents: plant resins (constituting 50–70%), beeswax (making up 30–50%), essential and aromatic oils (comprising 5–10%), and pollen (contributing 5–10%), in addition to trace quantities of organic compounds and minerals [18,45]. Out of these constituents, the resinous compounds form the bulk of the materials. Phenols and terpenes-class compounds are of especially great interest as these molecules are often associated with propolis biotherapeutic functionalities.

Phenols belong to a group of chemical compounds characterized by the presence of a hydroxyl group covalently bonded to an aromatic ring. These phenolic compounds are associated with a diverse array of biotherapeutic roles, including the inhibition of specific enzymes, antioxidative effects, stimulation of select hormones and neurotransmitters, and antimicrobial activities [78,79,80,81,82]. Propolis is known to contain various phenolic compounds, such as flavonoids, phenolic acids, tannins, stilbenes, curcuminoids, coumarins, and quinines [76,78,79,80,81,82]. 

Flavonoids constitute the predominant class of phenolic compounds present in propolis and are often linked to the material’s bioactivity. Their concentration frequently serves as a benchmark for gauging the quality of propolis. Brazilian propolis, renowned for its powerful biotherapeutic attributes, boasts substantial phenolic and flavonoid contents, reaching levels of 27.4% and 4.4%, respectively. In contrast to propolis from other South American countries, Brazilian propolis exhibits significantly higher active ingredient levels and thus a greater therapeutic potential [83]. A recent study by Hernandez Zarate et al. [84] characterized propolis from Guanajuato, Mexico. The Guanajuato propolis displayed an exceptional flavonoid content, with concentrations reaching as high as 379 mg of quercetin equivalents per gram of propolis. This stands as one of the most notable flavonoid contents documented, surpassing those of propolis from regions, such as China, Macedonia, Portugal, Argentina, Australia, Bulgaria, Chile, and even Brazil. Corresponding biofunctional assays yielded compatible outcomes, with Guanajuato propolis showcasing superior antioxidant characteristics.

Terpenes are naturally occurring compounds that can be found in a wide range of animals and plants. Due to their highly volatile nature, terpenes easily evaporate, releasing a pronounced scent and flavor that serves as a deterrent against pests. These hydrocarbon molecules are often responsible for the distinctive resinous aroma and contribute to certain pharmacological effects of propolis. Among the various types of terpenes, sesquiterpenes are the most prevalent class of terpene compounds discovered in propolis [3]. In Brazilian propolis, three specific sesquiterpenes (namely, *γ*-elemene, *α*-ylangene, and valencene) are present in notable concentrations (6.25%, 1.00%, and 1.25%, respectively). The effective antibacterial activity of Brazilian propolis is attributed to the presence of these organic compounds [85]. An in-depth study on Iranian propolis highlighted the bactericidal activity of sesquiterpenes and their contribution to the reported antimicrobial function of the propolis. Mono- and sesquiterpene alcohols were isolated using an ethanol extraction and tested against various microbes. Significant zones of inhibition were visually observed when the respective isolates were tested against *Staphylococcus aureus* [86]. Souza et al. [87] evaluated the chemical compositions as well as antioxidant and antimicrobial activities of propolis from both stingless bees (*Frieseomelitta longipes*) and honey bees in north Brazil. A thorough examination using gas chromatography-mass spectrometry (GC-MS) revealed the existence of 45 distinct sesquiterpene compounds in the propolis extracts under examination. Subsequent biofunctional assessments demonstrated considerable antioxidant and antimicrobial potential within the individual propolis extracts. These findings additionally confirmed the significance and contributions of terpenes in conferring bioactive properties to the propolis composition.

While phenols and terpenes are abundantly found in most propolis, their concentrations, proportions, and varieties often vary between propolis due to differences in both the extrinsic and intrinsic factors, such as botanical sources, geographical origin, extraction methodologies, climate, and bee species [88].

## 3. Production of Propolis by Bees

Propolis is a viscous and resinous complex, primarily composed of plant resin with beeswax, oil, pollen, and trace elements forming the remaining components [76]. A specialized group of bees, engaged in a task known as “cementing activity”, is responsible for the collection of resins and the subsequent processing to create the final propolis product [89]. The production of propolis commences with these cement bees scouting their surroundings to gather resinous substances from diverse botanical sources, such as birch, conifers, elm, palm, pine, poplars, willow, Asteraceae, coinvine, and horse chestnut trees [90]. The procedure of collecting resin can be condensed into eight steps (Figure 1).

Bees collect and extract sticky resin from plants using their mandibles. They then engage in manipulating the harvested resins by mixing them with salivary enzymes, subsequently transferring them to their forelegs. The processed resins are then transferred from the forelegs to either of the middle legs. Finally, the bees move the treated resins along and pack them into the corbicula on the same side. Subsequently, the cement bees transport the processed resins back to the hive, carrying them within the corbicula. These treated resins are promptly administered to the designated hive area. Once applied, the cement bees carefully form and refine the resins, eliminating any undesirable roughness. Only once a smooth resin patch is achieved, the cement bees move on to the subsequent stage, which involves the removal of wax from the comb and its integration into the resin materials culminating in the final propolis formation. This infusion of wax enhances the propolis texture, reducing its stickiness and rendering it more solid [89]. The precise procedure and sequential steps of propolis production may vary depending on the bee species and prevailing environmental factors (Figure 2).

Most stingless bees (meliponines) create propolis in the same way as honey bees. Several meliponines incorporate soil in their propolis, which increases the ultimate bulk and volume of the product [91].

## 4. Factors Affecting the Quality of Propolis

### 4.1. Botanical Effect on Propolis

As a major component of propolis, the botanical origins, geographical locations, and nature of the resins significantly impact the composition and quality of the propolis produced by bees [7,70,92]. This in turn affects the bioactivity of the resulting propolis extracts. Table 1 highlights some of the common compositional and functional properties that are reported in diverse propolis types. Table 2 shows the main phytochemicals and microbial activities that are reported in propolis types produced by diverse stingless bee species.

The diversity and complexity of propolis often complicate analyses, preventing the accurate determination of the respective resin sources. However, an extensive review of the literature illuminated some commonly targeted botanical species from which bees gather resin. Among these, *Populus* spp. stands out [155]. *Populus* spp. comprises a group of woody plants that are commonly distributed across temperate regions worldwide, including Europe, North America, West Asia, and New Zealand [18,74,156]. Elements of poplar resin have also been identified in propolis collected from diverse global locations, like Egypt, China, Korea, Croatia, Taiwan, and Africa [3,82,90]. A recent investigation conducted by Oryan, Alemzadeh and Moshiri [18] revealed substantial concentrations of valuable bioactive compounds, such as flavonoids, phenolic acids, and esters, in *Populus* spp. extracts. It is hypothesized that the incorporation of these bioactive compounds within propolis contributes to the advantageous biotherapeutic potential of this valuable bee product.

Among the varieties of propolis, Brazilian propolis has attracted significant interest due to its remarkable biotherapeutic properties. Brazilian propolis can be broadly classified into four distinct types: poplar propolis, brown propolis, green propolis, and red propolis. These types are said to originate from different botanical sources, specifically *Populus* (Salicaceae), *Hyptis divaricata*, *B. dracunculifolia*, and *D. ecastophyllum*, as outlined by Franchin et al. [157]. Regardless of the propolis types and botanical sources, the composition of propolis is directly related to that of bud exudates collected by bees. For example, Brazilian red propolis was recently discovered as the 13th group of Brazilian propolis. An analysis conducted by Rufatto et al. [99] emphasized the biotherapeutic properties and distinctive red identity of Brazilian red propolis. This type of propolis is predominantly located in the northeastern region of Brazil and is primarily sourced from *D. ecastophyllum*, a plant belonging to the Fabaceae family. A comprehensive examination of its composition revealed that Brazilian red propolis acquired a notable quantity of bioactive elements from the Fabaceae resins, thereby enhancing its exceptional biotherapeutic attributes [18,99]. Certain bioactive compounds found within this group encompass phenylpropanoids, terpenes, flavonoids, aromatic acids, and fatty acids. Furthermore, the propolis’ physical attributes are influenced by the plant source. The distinctive red color observed in Brazilian red propolis is linked to two flavanol pigments, Retusapurpurin B and Retusapurpurin A, abundantly present in Fabaceae resins, thus contributing to its coloration [18,99].

### 4.2. Bee Species Effect on Propolis

Each species of honey bee has certain characteristics that are unique to them. Every country has been races of local ecotypes, which are adapted to the ecological conditions of the region. Considering the process of propolis production, it is apparent that the differences in the ecotypes of bees have a profound effect on the physicochemical composition and functional properties of the propolis produced [3]. Studies on the effect of bee species on propolis are relatively scarce. A report by Silici and Kutluca [120] highlighted that there was a vast difference in the chemical composition of propolis garnered from different varieties of bees from the same apiary. Propolis from *A. mellifera anatoliaca* and *A. mellifera carnfica* contained a certain proportion of chrysophanol, naringenin, 2-propenoic acid, nonadecane, docosane, and vanillin. These compounds were absent in *A. mellifera caucasica* propolis. Resultantly, such a variation in the chemical composition directly influenced the bioactive functionalities of the respective propolis. In the same study, the authors showed that there was a significant difference in the antimicrobial potency and specificity between the respective propolis. *A. mellifera caucasica* propolis demonstrated greater potency against all tested microbes, with a lower minimum inhibitory concentration (MIC) value.

Comparable findings were noted regarding the propolis derived from stingless bees. Propolis generated by *M. scutellaris*, for instance, lacked both benzophenones and flavonoids in its composition chemical spectrum [5]. On the contrary, *M. fasciculata* geopropolis harvested from the same region contained high concentrations of polyphenols, flavonoids, triterpenoids, saponins, and even tannins [46]. On the same note, both Brazilian green and red propolis are produced by the same species of bee: Africanized *A. mellifera*. Despite their common origin, two types of propolis differ in chemical composition. Green propolis is rich in prenylated phenylpropanoids, whereas isoflavonoids are found in abundance in red propolis [69,158]. Moreover, these investigations underscored the considerable impact of bee species on the resulting propolis composition. One potential reason for these variations in composition can be linked to the bees’ preference for specific botanical sources. It is obvious that bees choose appropriate resin sources from different plant families according to their availability at a specific location and their suitability to their needs [159]. While Fabaceae has been reported as the preferred resin source amongst various bee species [160,161,162,163,164,165,166,167], other plant families, including Anacardiaceae [116,168], Apiaceae [86,169] and Asteraceae [158,170], are also harvested for propolis production. 

Leonhardt et al. [171] discovered that stingless bees employ comparable mechanisms and compounds for detecting and recognizing plant resin sources, mirroring the strategies honey bees use to locate and distinguish flowers. Stingless bees exhibit a strong tendency toward opportunistic resin collection, as all species gather resin from a similar range of tree species rather than relying on the entire scent blend. They identify and discern resin sources by evaluating various volatile mono- and sesquiterpenes. Moreover, there is a tendency among stingless bees to prefer familiar, extensively altered extracts, indicating a form of resin content recognition among different bee species and even within colonies. Among the factors influencing plant resin intake, predator attacks, like those from ants, has the most significant impact, while manual nest destruction only slightly increases the number of resin collectors. On the other hand, Popova et al. [28] reported that *Axestotrigona ferruaines* (cited as *Meliponula ferruginea* by Popova et al. [28]) gathered resin from various plants found near their nests, without displaying a specific preference for any single resin source. This can lead to propolis and cerumen displaying considerable variations in their chemical compositions. Salatnaya et al. [172] investigated forage plants for the stingless bees of the genus *Tetragonula* in west Halmahera, Indonesia, and revealed the utilization of seventy-seven distinct plant species for nectar, pollen, and occasional sources. Resins are necessary for nest construction by stingless bees. Although the study did not document bees specifically collecting resin from resin-producing plants, it identified nine resinous plant species in the collection areas that were likely to serve as suitable resin sources for these bees.

The ultimate question about how bees go about choosing their resin source, especially for the stingless bees, is whether they have a strong preference for the resin chosen by their species. This mystery remains unsolved among researchers. Another contributing factor to the variation in the chemical composition of propolis can be attributed to the diverse biological characteristics of the different bee species. As previously discussed, propolis includes a portion of beeswax produced by the bees. Consequently, unique enzymes and biochemical compounds specific to each bee species are introduced into the beeswax, which subsequently become part of the resulting propolis. These bioactive compounds can initiate specific biochemical reactions, leading to distinctions in the formed propolis. Further research is necessary to achieve a greater insight into the impact of these bioactive compounds of entomological origin on the composition of propolis, meliponine, geopropolis, and cerumen across various stingless bee species.

### 4.3. Effect of Extraction Processes on Propolis

Raw propolis harvested from beehives experiences a series of extraction processes before being presented to consumers. These extraction processes are necessary to isolate and concentrate bioactive compounds while at the same time sieving out non-functional bulk components (e.g., beeswax, pollens, and fibers) from the raw propolis. In order to preserve the unique structural properties and biotherapeutic functionalities of sensitive bioactive compounds, the extraction process should be conducted under mild aqueous conditions. However, conventional aqueous-based extraction strategies are not adopted for bee propolis due to the much lower efficiency of extraction associated with the use of water as an extraction medium [82,88,92,173]. Moreover, the lipophilic nature of these bioactive compounds further hinders the extraction efficacy, preventing solvation into aqueous media [7,92]. Instead, maceration with organic solvents is a widely employed strategy for extraction of propolis. Some popular solvent choices include ethanol [9,80,82,92], methanol [174], and propylene glycol [175]. A study by Silva et al. [176] demonstrated the difference in extraction efficiencies between different solvents. Hydroalcoholic extracted Braganca propolis possessed the highest propolis (277.17 ± 7.50 mg) and flavonoid (142.32 ± 4.52 mg) contents. Comparatively, methanolic and aqueous extracted Braganca propolis showed significantly lower bioactive compound concentrations. In fact, maceration with 70% ethanol is widely reported in numerous publications as the preferred means for the extraction of propolis [80,125,175,177]. Despite its prolific extraction capability, ethanol is rarely utilized for the industrial extraction of propolis due to the overwhelming residual flavor and adverse health effects associated with the frequent consumption of alcohol [44]. Traditionally, ethanol-infused products are often faced with great resistance, especially when such products are applied for biomedical and consumer care applications [177].

As an alternative, propylene glycol is generally preferred as the main extraction solvent for the preparation of commercial propolis due to its excellent biocompatibility with mammalian cells [178,179,180]. A report compared the difference in extraction yields using different extraction solvents. As expected, the aqueous propolis extract showed the lowest phenolic content (1,207.9 ± 27.6 μg/mL), followed by the polyethylene glycol aqueous propolis extract (2,149.5 ± 16.1 μg/mL). The ethanolic propolis extract possessed the highest total phenolic content (20,791.3 ± 2,320.9 μg/mL), almost 10-times higher than that of the polyethylene glycol propolis extract. A corresponding antioxidant functionality assay revealed that only polyethylene glycol and ethanolic propolis extracts demonstrated significant mitochondrial superoxide and total intracellular reactive oxygen species (ROS)-decreasing properties. Such intracellular antioxidant properties were absent from the aqueous propolis extract, where the total bioactive compound content was lower [175]. A study by Ramanauskienė et al. [181] showed corresponding results. Upon the exposure of 5% propolis to different extraction solvents, ethanol-extracted propolis carried the highest amount of phenolic compounds (175.6 ± 1.89 mg/g), superior to propylene glycol (118.6 ± 1.78 mg/g)- and water (19.6 ± 0.93 mg/g)-extracted propolis. While propylene glycol is generally regarded as safe and accepted for biomedical applications, the incorporation of such synthetic chemical reagents into products is usually resisted by the general public. With a heightened consciousness for health and an increased pursuit of natural products, there is an increasing demand to replace such chemical reagents with natural green solvents. 

To this end, much of the recent research focuses on searching for a suitable green solvent for the propolis extraction process. A study evaluated the use of olive oil and β-cyclodextrin as alternative solvents for propolis extraction. The results from the study revealed that olive oil-extracted propolis possessed the highest yield of 4-geranyloxyferulic acid and a moderate amount of other chemical compounds (e.g., ferulic acid, boropinic acid, umbelliferone, 7-isopentenyloxycoumarin, and auraptene). On the other hand, β-cyclodextrin proved the most efficient in the extraction of ferulic acid from raw Italian propolis [182]. A feasibility study was conducted using virgin coconut oil and olive oil as extraction solvents for *Trigona* propolis. Subsequent bioactivity testing with coconut oil and olive oil propolis extracts demonstrated positive antimicrobial functionalities, with significant zones of inhibition [183]. While further analytical studies on the respective propolis extracts are needed, this preliminary result indirectly suggests the effectiveness of such natural oil-based reagents as potential natural solvents for propolis extraction.

Traditionally, water, ethanol, and methanol have been commonly used as solvents for propolis extraction due to their effectiveness in extracting a wide range of bioactive compounds from propolis. These solvents have been widely accepted in the field of propolis research for many years [82,184]. While propylene glycol and olive oil are not as frequently used as ethanol or methanol for the extraction of propolis, some researchers have explored them as solvents for this purpose [92,181,185]. Honey brandy and mead have been utilized in the propolis extraction reported by Freitas et al. [186]. L-lactic acid is indeed considered an alternative solvent to ethanol for propolis extraction, and it has gained attention in recent years due to its eco-friendly nature and its ability to extract certain bioactive compounds from propolis effectively [187]. However, it is important to note that the choice of solvent can significantly impact the composition and properties of the resulting extract. In recent years, researchers have explored the use of new or advanced solvents for propolis extraction to enhance efficiency and target-specific bioactive compounds. Some of these advanced solvents include natural deep eutectic solvents (NADESs) [188,189], ionic liquids [190,191], and supercritical CO_2_ [192,193,194]. These solvents offer advantages in terms of selectivity, reduced environmental impact, and improved extraction yields for specific compounds.

However, extraction methods influence the quality and properties of propolis. The conventional and simplest method for utilizing the therapeutic potential of propolis involves the utilization of alcohol extraction. The process of extraction is of the utmost importance in accessing the bioactive constituents of propolis, and the careful selection of an appropriate extraction method is essential in guaranteeing the creation of propolis-based products that are both of a high quality and cost-effective. According to a study conducted by Pobiega et al. [195], the antibacterial activity of extracts, extraction yields, and the levels of phenolic and flavonoid components are influenced by different extraction methods. The utilization of ultrasound-assisted shaking extraction has been found to yield superior outcomes compared to traditional shaking extraction and ultrasound-assisted extraction methods. From traditional maceration extraction, Soxhlet extraction to advanced methodologies, such as ultrasound extraction (UE), microwave-assisted extraction (MAE), supercritical CO_2_ extraction, and high-pressure methods are explored. Furthermore, the choice of solvents is a critical consideration, where water–ethanol mixtures continue to demonstrate their efficacy, while oils and natural deep eutectic solvents (NADESs) exhibit promising potential for propolis extraction [196,197,198]. Notably, as per Bankova, Trusheva and Popova [196], ultrasound-assisted extraction emerges as the optimal method, balancing considerations, such as extraction time, yield, and cost-effectiveness.

## 5. Chronological Applications of Propolis

Propolis has been identified and used by humans as a folk medicine since 300 B.C. It is reputed to possess multiple biotherapeutic properties, such as anticancer, antioxidant, antimicrobial, antiviral, and immunomodulatory functions [12,55,88,124,199]. In ancient Egypt, propolis was essentially used as an antiputrefactive agent to embalm corpses [7]. Propolis was also adopted by the Persians, Greeks, Romans, and Incas in folk medicines to treat various maladies [38,106,199]. The physicians in Greek and Rome mainly used propolis as a mouth antiseptic and created poultices for wound therapy purposes [17]. In addition, it was also recorded in the Old Testament that “tzori”, a Hebrew word for propolis, was considered and used as a healing medicine [74]. The acceptance of propolis as a legal drug only occurred in the seventeenth century, when propolis was officially registered as a drug in the London pharmacopoeia [74,200]. The all-natural potent antibacterial activity of propolis made it a popular candidate in Europe from the seventeenth to twentieth centuries for treating inflamed wounds, internal ulcers, and excoriations [7,12]. In the nineteenth century, propolis was extensively used as a healing agent, especially during the Second World War, as a natural compound to treat tuberculosis [7]. The first patented scientific work on propolis was published in 1904 and, since then, there have been increasing publications on its characteristics and biological activities [7,70].

## 6. The Antimicrobial Component of Propolis

In the present day, due to its various biotherapeutic properties, propolis has been utilized for a variety of applications, including as a natural active ingredient in medicines, a functional nutraceutical product for nutritional needs, and a structuring material for consumer care products. One of the most prominent biofunctionalities of propolis lies in its bactericidal potency. As mankind steps into the post-antibiotics era, propolis is regarded as having the potential to step up as the future-generation antimicrobial agent of choice for treating microbial infections. The prevalence and increasing outbreaks of antimicrobial-resistant pathogen-based infections pose a global health issue. The gravity of the issue is compounded by the lack of further developments of new antibiotics that can effectively target and kill such resilient microbes [201,202,203,204]. As a result, there is a shift toward searching for alternative antimicrobial compounds with potent bactericidal capabilities. Natural products with antimicrobial activities stand out as potential candidates due to the greater chemical diversity and complexity that deter bacteria from gaining resistance [57]. Propolis has since attracted attention as a potential candidate and/or ingredient in antimicrobial drug development. In nature, propolis functions as a biocide, deterring pests and pathogens from invading the hive of honey bees. Despite the major variations in the chemical composition of propolis derived from different geographical and botanical origins, antimicrobial assays with different propolis have shown that their antibacterial potency remains relatively similar [205]. Propolis possesses noteworthy antibacterial properties. It exhibits efficacy against both Gram-positive and Gram-negative bacteria, as well as aerobic and anaerobic microorganisms. However, the efficacy of propolis is contingent upon its chemical composition and exhibits variations across different regions [206]. The antibacterial and antifungal activities were demonstrated to be similar for both Greek and Cypriot propolis ethanol extract [207]. Both were shown to effectively inhibit the proliferation of Gram-positive pathogens (*S. aureus*, *S. epidermidis*, *B. cereus*, and *L. monocytogenes*) and fungi. However, both propolis were inactive against several lactic acid bacteria (*Lactobacillus delbrueckii subsp. delbrueckii* and *Lactobacillus plantarum)*. In-depth studies have shown that propolis is more effective against single-walled Gram-positive microbes, as compared to double-walled Gram-negative bacteria. Serbian propolis is effective against both Gram-positive and Gram-negative bacteria [45]. It was predominantly effective against the Gram-positive strains *L. monocytogenes*, *B. subtilis*, *E. faecalis*, and *S. aureus*. In a corresponding study, Mahabala et al. [208] also showed that the propolis ethanol extract was able to inhibit the growth of the Gram-positive strains *S. mutans* and *Lactobacillus acidophilus* with both MICs at 100 mg/mL. Hydrophobic compounds, such as phenols, flavonoids, and terpenes, have been reported as chief bioactive compounds responsible for the observed antimicrobial activities. Ramanauskienė et al. [181] demonstrated the importance of phenolic compounds on the antimicrobial potency of propolis. Using a variety of aqueous and organic extraction solvents, the authors managed to vary the phenolic concentration extracted from raw propolis. Water extracts yielded the lowest phenolic content (14.4 ± 0.22 mg/g), while the ethanol extract was highest with a reported value of 167.50 ± 2.78 mg/g of phenolic compounds. Subsequent antimicrobial testing showed that water-extracted propolis possessed no bactericidal activities, whereas ethanol-extracted propolis showed a good antimicrobial capacity against a broad spectrum of microbes. Propolis extract from the Czech Republic possessed the highest phenolic content, amounting for a 129.83 ± 5.9 mg caffeic acid equivalent per gram of propolis. The antimicrobial assay demonstrated corresponding results, with the Czech propolis extract illustrating a potent bactericidal potency, having a minimum bactericidal concentration range of around 0.1 to 2.5 mg/mL against all fourteen Gram-positive microbes [209]. These studies further reinforced the importance of hydrophobic compounds, such as phenols, in imparting antimicrobial potency to propolis. While the exact antibacterial mechanism of phenolic compounds is yet to be fully deciphered, it is hypothesized that these compounds interact favorably with and insert themselves into the phospholipid cell membrane. As the concentration of phenols on the phospholipid bilayer increases, it adversely affects the membrane’s rigidity, eventually leading to the total loss of structural integrity, killing the microbes [210]. The membrane-targeting mode of bactericidal action of the phenol compounds can be one of the main factors attributed to the difference in the antimicrobial potency of propolis against Gram-positive and Gram-negative microbes. The presence of a double cytoplasmic membrane in Gram-negative microbes acts as an additional physical barrier, effectively deterring the hydrophobic molecules from penetrating and lysing the cell membrane [12,45,205,207,211]. While the antimicrobial potency of propolis has been well established, there exist many unknowns that are worth evaluating. A particular area of great scientific interest involves the identification of the chief bioactive compound in propolis responsible for the potent antimicrobial activity, as well as elucidating the exact mechanism of the bactericidal action of the natural material. Insights into these areas aid the progression of antimicrobial science and guide the development of next-generation antibacterial drugs.

## 7. Quality Control of Propolis

The chemical composition of propolis is influenced by the geographical location and seasonal variations, presenting a significant challenge to its standardization and quality control [212,213]. The gross composition, encompassing parameters, like total phenolic compounds, total flavonoids, waxes, ashes, moisture, and insoluble residue, serves as a fundamental basis for establishing the quality standards for propolis [214]. The extraction method and the selection of solvents are critical stages that not only determine the quality, but also impact the yield of bioactive components present in propolis [196,213,215,216]. 

Ensuring the quality of propolis involves a series of steps and precautions to meet the predetermined standards for purity, potency, and safety. Identifying the botanical and geographical origins offers insights into a plant’s composition and potential bioactivity. Techniques, such as pollen analysis and chemical profiling, authenticate the origin of propolis [217,218]. Physical and chemical analyses aid in detecting the impurities and unwanted substances. Analytical methods, such as high-performance liquid chromatography (HPLC), GC-MS, liquid chromatography mass spectrometry (LC-MS), mass spectrometry fingerprints, including electrospray ionization mass spectrometry (ESI-MS), and nuclear magnetic resonance spectroscopy (NMR), quantify the specific bioactive compounds [217]. HPLC with a gradient mode, coupled with photodiode array detection, remains the preferred technique for assessing major propolis components. Atmospheric pressure chemical-ionization mass spectrometry (APCI-IT-MS) offers an alternative method for obtaining characteristic propolis fingerprints and the reliable identification of numerous components [219]. Chromatographic methods segregate the intricate propolis matrix, enabling component isolation, identification, and quantification. While fingerprinting methods, like ESI-MS, assist in sample characterization and determining the plant origin, chromatography remains essential for compound quantification [55].

Distinctive chemical profiles characterize various propolis types, underscoring the infeasibility of a universal set of criteria for standardization. Instead, tailored criteria based on the concentrations of bioactive secondary metabolites must be established for specific propolis types. To address this, the International Honey Commission proposed concentration values for biologically active constituents [55,217]. Furthermore, microbial contamination can compromise propolis quality and safety. Microbiological testing assesses the presence of harmful microorganisms. Bogdanov [220] conducted a comprehensive review of potential contamination resulting from beekeeping practices and environmental factors. As a result, ensuring the safety and quality of propolis products requires a thorough consideration of contaminants originating from both beekeeping practices and the environment. This precaution is crucial to guarantee the suitability of propolis products as effective and safe alternative antimicrobial agents. Nonetheless, in the context of propolis quality control, it is imperative to conscientiously account for potential contamination originating from external sources, encompassing xenobiotics, pesticides, and toxic metals [221].

## 8. Conclusions

Propolis, geopropolis, and cerumen, sticky substances naturally created by honey bees and stingless bees as an adhesive sealant for their hives, have garnered interest as superfoods with remarkable biotherapeutic attributes. The traditional use of these bee products holds a significant place in various cultures due to their remarkable properties. Its historical applications range from wound healing and oral health to its potential role as an antimicrobial agent. While modern research has shed light on their diverse biotherapeutic potential, it is important to respect and acknowledge the rich traditional knowledge that has surrounded their use for centuries. Numerous internal and external factors are demonstrated to directly influence the physical and chemical characteristics as well as the overall quality of propolis. These factors encompass the source of botanical resins, the honey bee species involved, and the methods of extraction. Despite their heterogeneous composition and properties, propolis and propolis-like products have been extensively documented for their diverse therapeutic effects, which encompass potent antimicrobial capabilities, effective anti-cancer properties, and impressive immunomodulatory activity. While this bee glue shows promising potential as a future-generation therapeutic agent for addressing infections and health issues, it is important to exercise caution. Further research is imperative, particularly for unveiling the underlying mechanisms of its action and delving into the in vivo safety and efficacy of these natural compounds. While bee propolis is recognized for its natural antimicrobial properties and potential antibiotic alternatives, there is a promising application that requires continued research to further understand geopropolis and cerumen. Future research should disclose the potential and benefits of these natural bee-derived substances.

## Figures and Tables

**Figure 1 foods-12-03909-f001:**
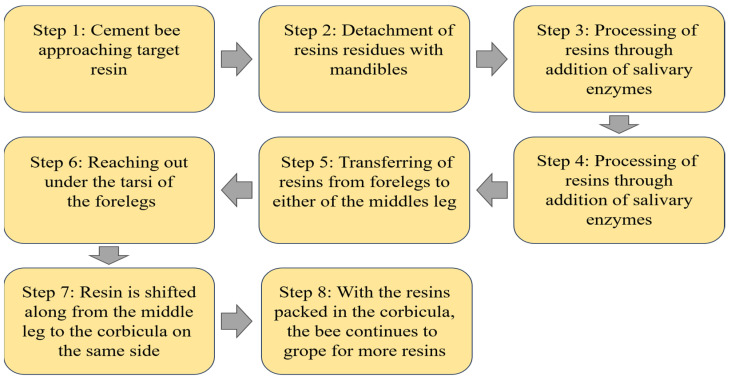
Resin harvesting process by cement bees [89].

**Figure 2 foods-12-03909-f002:**
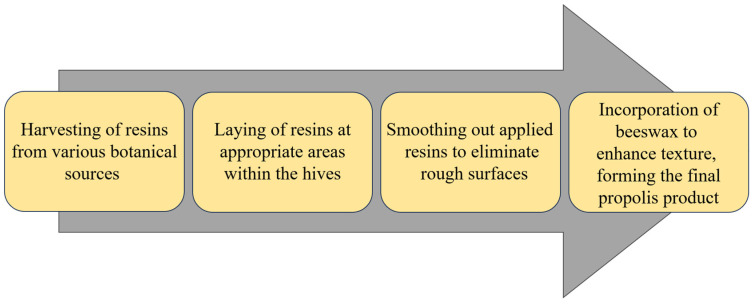
Overall process flow of propolis production by bees.

**Table 1 foods-12-03909-t001:** The chemical composition and functions of propolis from various botanical sources.

Type of Propolis	Plant Source	Main Composition	Function	Reference
Bulgarian	Unknown	Flavonoids and esters of caffeic and ferulic acids	Antimicrobial	[93]
Brazilian	*Baccharis* spp., Clusia minor, Clusia major, *Araucaria heterophylla*	Coniferyl aldehyde, betuletol, kaempferide, and ermanin	Cytotoxic to fibrosarcomas and carcinoma cells	[94]
Brazilian	Unknown	Cinnamic acids, phenolic acids, flavonoids, fatty acids, diterpenes, triterpenes, polyphenols, and phenolic lipids	Anticancer Antimicrobial	[95]
Brazillian green	* Baccharis dracunculifolia *	Caffeic acid, *p*-coumaric acid, drupanin, kaempferide (as kaempferol), artepillin C, total flavonoids as quercetin, and total phenol content as gallic acid	Antimicrobial	[96]
Brazilian green	*Baccharis dracunculifolia*	Artepillin C, baccharin, and drupanin	Anti-inflammatory	[97]
Bazillian green	Unknown	Caffeoylquinic acid derivatives	Angiostatic	[98]
Brazil red	*Dalbergia ecastophyllum*	Formononetin, biochanin A, liquiritigenin, and flavonoids	Antimicrobial	[99]
Chinese	Probably *Populus* spp.	Caffeic acid, benzyl caffeate, phenethyl caffeate, 5-methoxy pinobanksin, pinobanksin, pinocembrin, pinobanksin-3-O-acetate, chrysin, and galangin	Antioxidant	[82]
Chinese	Unknown	Caffeic acid, *p*-coumaric acid, ferulic acid, 3,4-dimethoxycinnamic acid, pinobanksin, cinnamylideneacetic acid, caffeic acid phenethyl ester, chrysin, pinocembrin, galangin, pinobanksin 3-acetate, cinnamyl caffeate, and tectochrysin	Antioxidant	[100]
Cyprus	*Pinus* spp., *Cedrus* spp., *Juniperus* spp., maquise trees, olive, carob trees	8-βH-cedran-8-ol	Antimicrobial	[101]
Egyptian	Unknown	Caffeic acid phenethyl ester (CAPE)	Antiviral	[102]
Ethiopian	Unknown	Betulinic acid	Antimicrobial	[103]
Ethiopian	Unknown	Saponins, tannins, flavonoids, steroids, triterpenes, and glycosides	Antimicrobial	[104]
Europe and Central Asia (Poland, Ukraine, Kazakhstan, Greece)	Unknown	p-Coumaric acid, chrysin, pinocembrin, sakuranetin, galangin, and pinobanksin-3O-acetate	Antimicrobial	[105]
Greece	Unknown	Pinocembrin, chrysin, galangin, apigenin, pinobanksin 3-O-acetate, and (±) catechin	Antioxidant	[106]
Greece	Unknown	Totarol, manoyl-oxide, ferruginol, epitorulosol, 13-epitorreferol, agathadiol, manool, copalol, 14,15-dinor-13-oxo-8(17)labden-19-oic acid, pimaric acid, imbricataloic acid, and 13-epi-cupressic acid	Antimicrobial Antioxidant	[107]
Indian	Unknown	Pinocembrin and galangin	Antioxidant	[108]
Indian	Unknown	3,3,4-trimethyl-4-p-tolyl, naphthalelone derivitives, nicotinic acid, 5-phenoxymethyl-1,3,4-thiadiazol-2-amine, acetate 3-cyclohexen-1-ol, boron (methanamine)tris(trifluromethyl), and 2-methyl,1-penten-3-yne	Antimicrobial	[109]
Indonesia	*Calophyllum inophyllum*	Chromanone derivative and calophylloidic acid A	Antimicrobial	[110]
Iranian	*Populus* spp.	Pinobanksin, pinobanksin-3-acetate, pinocembrin, pinostrobin, and flavones, like chrysin and galangin	Antimicrobial	[111]
Kazakhstan	Unknown	Pinocembrin, galangin, pinobanksin and pinobanskin-3-O-acetate, and caffeic acid phenethyl ester	Antimicrobial	[112]
Korean	Unknown	Caffeic acid, *p*-coumaric acid, 3,4-dimethoxycinnamic acid, apigenin, kaempferol, pinobanksin, cinnamylideneacetic acid, chrysin, pinocembrin, galangin, pinobanksin 3-acetate, phenethyl caffeate, cinnamyl caffeate, and tectochrysin	Antioxidant	[113]
Lithuania	Unknown	Ferulic, caffeic, and *p*-coumaric acids	Antimicrobial Antioxidant	[92]
Malaysian	Unknown	Phorbol, isolongifolol, germacrene D, isoaromadendrene epoxide, *α*-eudesmol, propanoic, octadecatrienoic acids, ribitol, arabitol, arabinitol, and D-glucitol	Antioxidant	[114]
Malaysian	Unknown	3′-–*O*-methyldiplacone, nymphaeol A, and 5,7,3′,4′-tetrahydroxy-6-geranyl flavonol	Antioxidant Anti-inflammatory Anti-acne	[115]
Myanmar	Unknown	(22Z,24E)-3-oxocycloart-22,24-dien-26-oic acid	Cytotoxicity against human pancreatic cancer cell line	[116]
New Zealand	Unknown	Caffeic acid phenethyl ester	Antiviral	[117]
Nepal	Unknown	2’-Hydroxyformononetin, odoratin, 2-(1-Phenylprop-2-enyl)benzene-1,4-diol, vestitol (2’,7-dihydroxy-5-methoxyisoflawan), butein, dalbergin, 7-Hydroxyflavanone, and pinocembrin	Antimicrobial	[118]
Poland	Unknown	Chrysin, caffeic acid, *p*-coumaric acid, and ferulic acid	Anticancer	[119]
Portugal	Unknown	Chrysin, caffeic acid isoprenyl ester, and pinocembrin	Antimicrobial Antioxidant	[120]
Portugal	*Cistus ladanifer*, *Arbutus unedo*, *Lavandula stoechas*, *Thymus serpyllum*, *Eucalyptus* sp.	Pinobanksin, chrysin, acacetin, apigenin, pinocembrin, and kaempferol-dimethyl-ether	Antimicrobial	[121]
Romanian	Unknown	Chrysin, ferulic acid, galangin, *p*-coumalic acid, pinocembin, and quercetin	Antimicrobial Antioxidant	[122]
Saudi Arabia	Unknown	4-methyl salicylic acid, cinnamic acid, chrysin, gallic acid, apigenin, and myricetin	Antimicrobial, Antioxidant	[123]
Sonoran	*Populus* spp.	Pinocembrin, pinobanksin 3-acetate, chrysin, CAPE, acacetin, and galangin	Antioxidant Antiproliferative	[124]
South	Unknown	Gallic acid, caffeic acid, coumaric acid, artepillin C, and pinocembrin.	Antimicrobial Antioxidant	[12]
Taiwanese green	*Macaranga tanarius*	Propolins C, D, F, and G	Antimicrobial	[125]
Thai	Unknown	Rutin, quercetin, and naringenin	Antimicrobial Antioxidant	[126]
Thai	Unknown	Cardols, carnadols, anacardic acids, and triterpenes	Antimicrobial	[127]
Turkish	Unknown	Caffeic acid phenethyl ester (CAPE), galangin, chrysin, dimethoxycinnamic acid, and caffeic acid	Antiviral	[128]
Vietnamese (stingless bee)	Unknown	23-hydroxyisomangiferolic acid and 27-hydroxymangiferolic acid	Cytotoxicity against PANC-1 human pancreatic cancer cell line	[129]

**Table 2 foods-12-03909-t002:** The chemical composition and antimicrobial activity of stingless bee propolis and geopropolis from various botanical and entomological sources.

Geographical	Stingless Bee Species	Main Composition	Antimicrobial Activity Against	Reference
Argentina	*Scaptotrigona aff. postica*, *Tetragona clavipes*, *Melipona quadrifasciata quadrifasciata*, *Tetragonisca fiebrigi*	Diterpenoids, triterpenoids, resorcinols, salicylates	*Bacillus cereus*, *Bacillus subtilis*, *Candida albicans*, *Escherichia coli*, *Paenibacillus larvae*, *Pseudomonas aeruginosa*, *Staphylococcus aureus*	[130]
Australia	*Tetragonula carbonaria*	*C*-methyl flavanones, phloroglucinols	*P. aeruginosa*, *S. aureus*	[131,132]
Brazil	*Frieseomelitta longipes*	Monoterpenes, sesquiterpenes	*B. cereus*, *C. albicans*, *C. tropicalis*, *E. coli*, *P. aeruginosa*, *S. aureus*	[87]
Brazil	*Melipona fasciculata*	Flavonoid, hydroalcoholic	*C. albicans*, *Streptococcus mutans*	[1]
Brazil	*Melipona fasciculata*	Benzoic acid, dihydrocinnamic acid, coumaric acid, caffeic acid, prenyl-p-coumaric acid, flavonoids, artepillin C, trihydroxymethoxy flavonon, tetrahydroxy flavonon, triterpenes	*Pythium insidiosum*	[133]
Brazil	*Melipona fasciculata*	Ethanolic extract	*Actinomyces naeslundii* m104, *Enterococcus faecalis* ATCC 29212, *P. aeruginosa* ATCC 25619, *S. aureus* ATCC 25923 MRSA, *S. mutans* UA 159	[1]
Brazil	*Melipona quadrifasciata anthidioides*	*Ent*-kaurene diterpenoids, kaurenoic acid	*S. aureus*	[134]
Brazil	*Melipona quadrifasciata anthidioides*	Di- and trigalloyl and phenylpropanyl heteroside derivatives, flavanones, diterpenes, triterpenes	Gram-positive bacteria, Gram-negative bacteria, yeasts	[50]
Brazil	*Melipona quadrifasciata anthidioides*, *Scaptotrigona depilis*	Ethanolic extracts	Vancomycin-resistant *Enterococcus* (VRE) *faecalis*	[135]
Brazil	*Melipona quadrifasciata quadrifasciata*, *Tetragonisca angustula*	Flavonoids, terpenes as major constituents	*E. faecalis*, *E. coli*, *Klebsiella pneumoniae*, *Methicillin-resistant Staphylococcus aureus* (MRSA)	[136]
Brazil	*Melipona orbignyi*	Polyphenol, flavonoid	*C. albicans*, *S. aureus*	[42]
Brazil	*Melipona scutellaris*	Ethanolic extract	*S. aureus*, *S. mutans*, MRSA strains	[5]
Brazil	*Scaptotrigona* aff. *postica*	Ethanolic extract	*B. megaterium*, *C. albicans*, *C. krusei*, *C. grabata*, *C. parapsilosis*, *C. guilliermondii*, *C. tropicallis* *E. coli* D31-resistant streptomycin, *Micrococcus luteus* *S. aureus*, *S. typhimurium*	[16]
Brazil	*Scaptotrigona bipunctata* *Melipona quadrifasciata* *Plebeia remota*	Ethanolic extract	*E. faecalis*, *E. coli*, *K. pneumoniae*, *Methicillin-resistant Staphylococcus aureus* (MRSA), *S. aureus*	[95]
Brazil	*Tetragonisca fiebrigi*	Phenolic compounds, alcohol, terpenes	*B.subtilis*, *E. faecalis*, *E. coli*, *K. pneumoniae*, *Proteus mirabilis*, *P. aeruginosa*, *S. aureus*, *S. epidermidis*	[42]
Brunei Darussalam	*Geniotrigona thoracica*, *Heterotrigona itama*, *Tetrigona binghami*	Flavonoids, phenolic acids, terpenes, aromatic acids	*S. aureus*, *P. aeruginosa*	[137]
Brunei Darussalam	*Heterotrigona itama*	Ethanolic extrtact	*B. subtilis*, *E. coli*, *P. aeruginosa*, *S. aureus*	[138]
Brunei Darussalam	*Geniotrigona thoracica*, *Heterotrigona itama*, *Trigona binghami*	Ethanolic extract, water extract	*B. subtilis*, *E. coli*, *P. aeruginosa*, *S. aureus*	[139]
India	*Tetragonula iridipennis*	Flavonoids, phenolics	*Aeromonas* spp., *Bacillus* spp., *E. coli*, *Klebsiella* spp., *Proteus* spp., *Salmonella* spp., *Staphylococcus* spp., *Vibrio* spp.	[140]
India	*Tretragonula* sp.	Ethanolic extract	*Acinetobacter baumannii*, *B. subtilis* ATCC 6633, *E. coli* ATCC 117, *K. pneumoniae*, *S. typhimurium* ATCC 23564, *S. abony* NCTC 6017 *S. aureus* ATCC 6538, *S. epidermidis* ATCC 1228, *S. schleiferi*, *S. pyogenes*	[109]
Indonesia	*Tetragonula fuscobalteata*	Ethanolic extract	*E. coli*, *S. aureus*	[110]
Malaysia	*Heterotrigona itama*	Ethanolic extract	*S. aureus*	[141]
Malaysia	*Heterotrigona itama*, *Geniotrigona thoracica*	Phenolics, flavonoids	*B. subtilis*, *E. faecalis*, *Listeria monocytogenes*, *S. aureus*	[142]
Malaysia	*Tetragonula biroi*	Methanolic extract	*Propionibacterium acnes*	[115]
Malaysia	*Heterotrigona itama*	Ethanolic extract	*E. coli*, *P. aeruginosa*, *S. aureus*	[143]
Malaysia	*Heterotrigona itama*, *Geniotrigona thoracica*	Methanolic extract	*S. aureus*	[144]
Mexico	*Melipona beecheii*	Phenolics, flavonoids, flavanones, dihydroflavonols	*C. albicans*	[145]
Mexico	*Melipona beecheii*	Phenolic compound, flavonoid	*Salmonella typhi*, *S. aureus*	[146]
Nigeria	*Dactylurina studingeri*	Ethanolic extract	*E. coli*, *Klebsiella* sp., *P. aeruginosa*, *S. aureus*	[147]
Tanzania	*Axestotrigona ferruginea* ^1^	Diterpenes, cardanol C17:1, resorcinols, anarcardic acids, quinic acid, caffeoylquinic acids, triterpenes	*C. albicans* ATCC 10239, *E. faecalis* ATCC 29212, *E. coli* ATCC 25922, *L. monocytogenes* ATCC 7644, *P. aeruginosa* ATCC 27853, *S. typhi* ATCC 14028, *S. aureus* ATCC 25923	[28]
Thailand	*Tetragonula laeviceps*	Water and methanolic extract	*Aspergillus niger*, *C. albicans*, *E. coli*, *S. aureus*	[56]
Thailand	*Tetragonula laeviceps*, *Tetrigona melanoleuca*	*T. laeviceps*: *α*-mangostin, mangostanin, 8-deoxygartanin, gartanin, *γ*-mangostin, garcinone, dipterocarpol, methylpinoresinol *T. melanoleuca*: 3-O-acetyl ursolic acid, dipterocarpol, ocotillone I, ocotillone II, mixtures of ursolic and oleanolic aldehydes, cabralealactones	*B. cereus*, *L. monocytogenes*, *Micrococcus luteus*, *S. aureus*, *S. epidermidis*, *S. pyogenes*, MRSA strains *E. coli*, *P. aeruginosa*, *S. aureus*, *Serratia marcescens*, *Salmonella typhimurium*	[39]
Thailand	*Tetragonula laeviceps*, *Tetrigona melanoleuca*	Phenolics and flavonoids, gallic acid, pinocembrin, quercetin	*Cryptococcus neoformans*	[148]
Thailand	*Tetragonula pagdeni*	Ethanolic extract	*E. coli* ATCC 25922, *S. aureus* ATCC 25923	[149]
The Philippines	*Tetragonula biroi*	Ethanolic extract	*E. coli*, *S. aureus*	[150]
Vietnam	*Lisotrigona cacciae*	Alk(en)ylresorcinols, anacardic acids, triterpenes, flavonoids, xanthones, other phenols, fatty acids	*C. albicans*, *E. coli*, *S. aureus*	[151]
Vietnam	*Lisotrigona furva*	Cycloartenone, cycloartenol, (24E)-3β-hydroxycycloart-24-en-26-al, mangiferonic acid, mangiferolic acid	*B. cereus*, *C. albicans*, *P. aeruginosa*, *S. aureus*	[152]
Vietnam	*Homotrigona apicalis*	Spathulenol, triterpenes, xanthones	*B. cereus*, *C. albicans*, *E. coli*, *L. fermentum*, *P. aeruginosa*, *S. aureus*, *Salmonella enterica*	[153]
Different locations in tropics and the temperate zone	*Melipona quadrifasciata*, *Melipona anthidioides*	Flavonoids, esters of phenolic acids	*C. albicans*, *E. coli*, *S. aureus*	[154]

^1^ Cited as *Meliponula ferruginea* in [28].

## Data Availability

The data used to support the findings of this study can be made available by the corresponding author upon request.

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
