# Peer review of "Exploring the Functional Properties of Propolis, Geopropolis, and Cerumen, with a Special Emphasis on Their Antimicrobial Effects"

_foods, 2023, doi:10.3390/foods12213909_

Round 1

Reviewer 1 Report

The review is based on the chemical and biological properties and characteristics of propolis. Although the review is very well written and contains mostly updated references, I find two major problems: first, the lack of originality. In the last 5 years there are more than 500 reviews about propolis and its characterization. On the other hand, the title does not correspond to the content of the review, since geopropolis and cerumen are mentioned but 80% of the review is only about propolis. On the other hand, it deals with the antimicrobial activity of propolis, but this is not the central topic of the review but another item in the review. I think this review could be substantially improved if it goes into more detail about geopropolis and cerumen. As it is written, the title is not appropriate and the review lacks of originality.

Reviewer 2 Report

The manuscript is interesting and the data are well presented. The research can be used for biomedical and consumer care applications.

However, I recommend the authors to update the reference list. Over 80% of references are more than 5 years old. Please update them.

 Minor editing of English language required.

Reviewer 3 Report

This review summarizes in detail the whole process of propolis and its antimicrobial active components from production to practical application, and analyzes the factors affecting its quality and biological activity. This review will be very helpful for the future development and research of propolis. However, there are a number of issues in the article that need to be addressed, including but not limited to those listed below. In conclusion, this manuscript is not suitable for publication at this time.

Main comments

1. There are obvious problems in Figure 2.

2. Please highlight in 4.3 the comparison and advantages of the new extraction solvents over the traditional ones?

3. The conclusion needs to be more detailed to highlight the main points and future directions of the article.

4. In item 7, please highlight the specific results of the effect of microorganisms on the quality of propolis and the latest research progress.

Minor errors

The stereo configuration E/Z should be italicized in all compounds.

It is OK.

Round 2

Reviewer 1 Report

Dear Authors

 I have also used Scopus for looking references about propolis, and found the numbers previosuly discussed.

I also find the following sentence used by the authors disrespectful. Authors cannot please a reviewer by inventing references that do not exist. This reviewer is not asking authors to invent references and my comment was probably not understood. I still strongly believe you cannot make a review on something that is very scarce, except to say that the bibliography is scarce.

Therefore, the authors are planning to publish a review including in it that references on this topic are scarce and therefore they cannot carry out an appropriate review. Unfortunately I do not agree on that point.

If the authors consider including a mention of these samples in the review, it can be accepted but the title should be modified.

In addition, the authors must modify the title by removing importance to the antimicrobial activity (like the suggestions made by the authors but eliminating cerumen and geopropolis). Finally, the authors add extra information where they establish that at the APimondia 2023 congress it was observed that the activity of DPPH It does not correlate with the polyphenol profile. It is widely known that colormorimetric methods ti evaluate antioxidant activity are not appropriate due their lack of specificity. Furthermore, several journals (some important journals such as FOOD CHEMISTRY) do not accept manuscripts in which the antioxidant activity is measured only by colorimetric methods and even less when only one is used. It is known that these methods serve as a guide or for standardization, but the antioxidant capacity must be evaluated by cell culture techniques.

Furthermore this last paragraph added by authors point out the need for more studies on this types of samples before a proper review can be published.

Author Response

Dear Reviewer,
We are pleased to inform you that we have carefully addressed each of your comments and suggestions in a detailed response document, which is attached to this system. In the response document, you will find a point-by-point explanation in response to your feedback.
Thank you very much
